# Role of Perilesional Sampling of Patients Undergoing Fusion Prostate Biopsies

**DOI:** 10.3390/life13081719

**Published:** 2023-08-10

**Authors:** Riccardo Lombardo, Giorgia Tema, Antonio Nacchia, Elisa Mancini, Sara Franco, Filippo Zammitti, Antonio Franco, Hannes Cash, Carmen Gravina, Alessio Guidotti, Giacomo Gallo, Nicola Ghezzo, Antonio Cicione, Andrea Tubaro, Riccardo Autorino, Cosimo De Nunzio

**Affiliations:** 1Ospedale Sant’Andrea, Sapienza University of Rome, 00185 Rome, Italy; rlombardo@me.com (R.L.); giorgiat88@hotmail.com (G.T.); antonio.nacchia@uniroma1.it (A.N.); elisa.mancini@uniroma1.it (E.M.); sara.franco@uniroma1.it (S.F.); filippo.zammitti@uniroma1.it (F.Z.); antonio.franco@uniroma1.it (A.F.); carmen.gravina@uniroma1.it (C.G.); alessio.guidotti@uniroma1.it (A.G.); giacomo.gallo@uniroma1.it (G.G.); nicola.ghezzo@uniroma1.it (N.G.); acicione@libero.it (A.C.); andrea.tubaro@mac.com (A.T.); 2Department of Urology, University of Magdeburg, 39106 Magdeburg, Germany; hannes.cash@prouro.di; 3Department of Urology, University of Chicago, Chicago, IL 60637, USA; ricautor@gmail.com

**Keywords:** prostate biopsy, perilesional sampling, targeted biopsies

## Abstract

Recently, researchers have proposed perilesional sampling during prostate biopsies to avoid systematic biopsies of patients at risk of prostate cancer. The aim of our study is to evaluate the role of perilesional sampling to avoid systematic biopsies of patients undergoing fusion biopsies. A prospective cohort of patients undergoing transrectal MRI transrectal fusion biopsies were consecutively enrolled. All the patients underwent systematic biopsies (SB), targeted biopsies (TB) and perilesional biopsies within 10 mm from the lesion (PB). The detection rates of different strategies were determined. A total of 262 patients were enrolled. The median age of those enrolled was 70 years. The mean BMI was 27 kg/m^2^, and the mean and prostate volume was 52 mL. A PIRADS score ≥ 4 was recorded in 163/262 (40%) patients. Overall, the detection rates of cancer were 43.5% (114/262) and 35% (92/262) for csPCa. The use of the target + peri-target strategy resulted in a detection of 32.8% (86/262) of cancer cases and of 29% (76/262) of csPCa cases (Grade Group > 2). Using the target plus peri-target approach resulted in us missing 18/262 (7%) of the csPCa cases, avoiding the diagnosis of 8/262 (3%) of nsPCa cases. A biopsy strategy including lesional and perilesional sampling could avoid unnecessary prostate biopsies. However, the risk of missing significant cancers is present. Future studies should assess the cost–benefit relationship of different strategies.

## 1. Introduction

Prostate biopsies represent the cornerstone of prostate cancer (PCa) diagnoses, with over one million procedures having been performed in Europe and in the United States [1,2]. Prostate biopsies are usually performed under local anesthesia either transrectally or transperineally [3,4]. Although several studies have demonstrated the superiority of the trans-perineal route over the transrectal route in terms of the detection rate and complications, the transrectal route is still the most widely used one [5].

The introduction of multiparametric MRI (mpMRI) in the diagnosis of PCa has dramatically improved the detection rate of clinically significant cancers [6,7,8]. A recent meta-analysis of 17 studies, including men who were suspected or confirmed to have PCa, evaluated the discrimination abilities of mpMRI. According to the meta-analysis, the average positive predictive values (PPVs) for significant PCa (ISUP grade > 2) in lesions with a PI-RADSv2.1 score of 3, 4 and 5 were 16% (ranging from 7% to 27%), 59% (ranging from 39% to 78%) and 85% (ranging from 73% to 94%), respectively [9]. However, there was considerable variation among the studies, indicating significant heterogeneity. In recent years, several strategies to determine the index lesion have been described, such as: fusion biopsies, in-bore biopsies and cognitive biopsies using either the transrectal or the trans-perineal approach [10]. Notwithstanding the vast amount of literature, there is no evidence to support one technique over the other considering that their detection rates of clinically significant cancers are comparable.

Ideally clinicians should only biopsy the target lesion, as in other tumors. However, the multifocality of PCa limits this approach. Using only targeted biopsies would miss 16% of ISUP > 2 and 18% of ISUP > 3 PCa cases. In biopsy-naïve patients, the best strategy remains the use of systematic + targeted biopsies, as recommended in the literature [11] and latest EAU guidelines [1,12]. Systematic biopsies refer to the sampling of 5–6 random cores per lobe and was considered the standard for many years before the introduction of mpMRI. Such a strategy has the aim of sampling different areas of the prostate. Target biopsies refers to the sampling of the mpMRI index lesion with 3–6 cores using a cognitive or fusion approach. Peri-targeted biopsies are defined as core samples of the surrounding area of the lesion within different radii from the lesion. Such a strategy may detect cancer cells in the surrounding area of the lesion [13].

The systematic + targeted approach is limited by there being a high number of clinically insignificant cancers diagnosed, a high number of cores needed (15–20 cores) and a consequently higher risk of complications. In order to overcome these limitations, some authors have proposed the sampling of the lesion and the perilesional area as an alternative to the abovementioned approach [14]. Although there are some interesting results, data are still scarce and retrospective, and prospective trials are needed to answer the question.

With this knowledge in mind, the aim of our study is to evaluate the role of perilesional sampling of patients at an increased risk of PCa undergoing transrectal fusion prostate biopsies.

## 2. Materials and Methods

Between September 2022 and April 2023, a consecutive series of patients in our prostate clinic with an abnormal MRI (PIRADS ≥ 3) were scheduled for a fusion transrectal ultrasound-guided prostate biopsy. Every patient provided explicit informed consent, and all procedures gained approval from the Local Ethics Committee. All the procedures were performed following the guidelines outlined in the Declaration of Helsinki. All patients with a previous history of prostate cancer or prostate surgery, PSA > 20 ng/mL, PIRADS score < 3 and those with previous negative biopsies were excluded from the study.

All patients with a detailed medical history were evaluated via a physical examination. DRE was performed by a senior staff urologist. Prostate volume was evaluated via a transrectal ultrasound and calculated using the ellipsoid formula. PSA was evaluated on the day of biopsy and PSA density was calculated (PSA/volume).

All mpMRI exams were reviewed by expert uro-radiologists with more than 5 years of experience in prostate MRI. In all cases, the number of suspicious lesions was reported, and the score was recorded according to PI-RADS v2 criteria.

Each patient in the study underwent a fusion prostatic biopsy under local anesthesia [15] based on the results of the mpMRI (BK Medical) [16]. The procedure was performed by experienced urologists who perform more than 100 procedures annually. The patients received a total of 12 systematic random biopsies (SBx), 3 biopsies per identified lesion (TBx) and 3 perilesional biopsies (PBx). Perilesional biopsies were taken within a 10 mm radius.

All biopsies were analyzed by a single dedicated uro-pathologist using the Epstein grading system [16,17]. In terms of disease classification, ISUP 1 was considered as a not significant disease (nsPCa), while ISUP ≥ 2 was classified as csPCa.

Complications were recorded according to the Clavien classification system [18].

### Statistical Analysis

Categorical variables are presented as frequency distributions and percentages, while continuous variables are represented by medians along with interquartile ranges (IQRs). To assess the detection rates of not significant prostate cancer (nsPCa) and clinically significant prostate cancer (csPCa), the approaches of a targeted biopsy (TBx) and a TBx with a perilesional biopsy (PBx) were compared to the detection achieved using a TBx with systematic SBx, which is considered as the “gold standard”. Sensitivities and their corresponding 95% confidence intervals (CIs) were calculated using binomial tests. Statistical significance was determined using non-overlapping 95% CIs between the sensitivity values.

Subgroup analysis was conducted based on the age, PSA density and PI-RADS assessment category. Differences in the detection rates between subgroups were evaluated using the Fisher’s exact test with the Freeman–Halton extension. All statistical analyses were performed using SPSS for MacOS, version 27 (IBM, Armonk, NY, USA).

## 3. Results

Overall, 262 patients with a median age of 70 years were prospectively enrolled. Overall, three patients presented with acute urinary retention after the biopsy, there were no post-biopsy fever cases, and no major complications were recorded. The baseline characteristics of the cohort are described in Table 1.

Overall, the detection rates of cancer were 43.5% (114/262) and 35% (92/262) for csPCa. The use of the target + peri-target strategy resulted in the detection of 32.8% (86/262) of cancer cases and of 29% (76/262) of csPCa cases (Figure 1). More specifically, systematic biopsies missed 10/262 (4%) of the clinically significant cancers, avoiding the diagnosis of 8/262 (3%) of the not significant cancers. The systematic biopsies plus target biopsies missed 2/262 (1%) of the csPCa cases, avoiding the diagnosis of 4/262 (1.5%) of the nsPCa cases. The target plus peri-target approach missed 18/262 (7%) of the csPCa cases, avoiding the diagnosis of 8/262 (3%) of the nsPCa cases. The target approach missed 22/262 (8%) of the csPCa cases, avoiding the diagnosis of 14/262 (5%) of the nsPCa cases (Table 2 and Table 3).

Overall, 163 patients presented with PIRADS 4-5. Using only SBx, a total of 10/163 (6%) csCancer and 6/163 (3.6%) nsCancer cases would have been missed. Using only SBx and TBx, a total of 2/163 (1%) csCancer and 2/163 (1%) nsCancer cases would have been missed. Using only target + peri-targeted biopsies, a total of 8/163 (5%) csCancer and 6/163 (3.6%) nsCancer cases would have been missed. Using only TBx, a total of 14/163 (5%) csCancer and 10/163 (6%) nsCancer cases would have been missed (Table 4).

Overall, 35 patients presented with PIRADS 5. Using only SBx, a total of 0/35 (0%) csCancer and 1/35 (3%) nsCancer cases would have been missed. Using only SBx and TBX, a total of 0/35 (0%) csCancer and 0/35 (0%) nsCancer cases would have been missed. Using only TBX + PBx, a total of 2/35 (6%) csCancer and 2/35 (6%) nsCancer cases would have been missed. Using only TBx, a total of 2/35 (6%) csCancer and 2/35 (6%) nsCancer cases would have been missed (Table 5).

## 4. Discussion

The present study adds important evidence to the role of perilesional sampling in prostate cancer diagnoses. According to our results, a systematic + perilesional biopsy technique carries the risk of missing 7% of the significant cancers. Moreover, for PIRADS ≥ 4 lesions, the risk is only 5%. Our results suggest that the TBx + PBx strategy may be suboptimal in patients undergoing fusion biopsies, considering the important risk of missing significant cancers.

In recent years, several authors have focused on this topic. However, most of the studies are retrospective, warranting prospective data to give a definitive answer to the question whether systematic biopsies may be avoided [19,20]. Hagens et al. evaluated 235 men undergoing fusion biopsies and observed the detection of csPCa in 92/235 (39%) using a TBx + PBx approach, with the risk of missing 3/235 (1%) of csPCa cases. Their encouraging results are severely biased by the low number of PIRADS 3 lesions included in the study, which limits the generalization of their results to other cohorts [10]. Additionally, the IQR of the systemic biopsies is wide (5–10), and the authors performed a high number of biopsies per lesion (five cores). Recently, Noujelm et al. evaluated 505 patient undergoing fusion biopsies, retrospectively evaluating the role of perilesional sampling based on the distance from the target. In their study, the TBx + PBx approach within a 10 mm radius resulted in the detection of 92% of csPCa cases, which is in line with our results of 93% of diagnosed csPCa cases. Additionally, the authors highlighted a different distribution depending on the PIRADS and PSA density. More specifically, the stratification into three risk groups based on the PI-RADS score and the PSAd might aid in the selection of patients for whom sampling beyond 10 mm can be safely omitted [21]. A sub-analysis to evaluate the role of peri-lesional sampling in patients older than 70 years old and in patients with a PSA density > 0.15 was performed. In both groups of patients younger and older than 70 years, the T + PT strategy missed 5% and 6% of the clinically significant cancer cases (*p* > 0.05). Additionally, in both group of patients with a PSA density >0.15 or >0.15, the T + PT strategy missed 7% and 7% of the clinically significant cancer cases (*p* > 0.05). This sub-analysis confirms the fact that the T + PT strategy is to be considered to be suboptimal for the diagnosis of PCa, even in different subgroups of patients.

The results of our study, together with the abovementioned experiences, should be interpreted with caution. The introduction of mpMRI in clinical practice to improve the detection of csPCa represents a cornerstone of prostate cancer diagnosis. In 2019, an MRI-first study by Rouviere et al. demonstrated that TBx + SBx detected 37% of csPCa cases, while SBx alone detected 29.5% of csPCa cases (*p* < 0.05) [22]. Additionally, in 2021, Preisser et al. observed a detection rate of 51.6% of csPCa for TBx + SBx vs. 45.7% for TBx alone [23]. Considering that a difference of 6–7% made TBx + SBx the gold standard approach, missing 5–7% of csPCa cases using only TBx + PBx is not acceptable nowadays. It is true that a TBx + SBx approach saves 12 cores per patient and allows a reduction of 3% of nsPCa diagnoses. However, the price to pay is probably too high particularly considering that no major complications were recorded when increasing the number of cores (only three AUR with a median of 20 cores per patient).

Another important role of perilesional sampling has been evaluated by Diamand et al. in reducing the upgrading events in the final pathology [24]. The authors enrolled 134 patients undergoing radical prostatectomy and concluded that perilesional sampling reduced the number of upgrading events in the final pathology when compared to those of systematic biopsies or targeted biopsies alone. Additionally, they observed similar discrimination abilities when comparing them to those of the gold standard (SBx + TBx). Although our study was not designed for this purpose, in our study, PBx detected a higher number of cancer cases (7/114 (6%) patients) when compared to that of TBx. These findings suggest the possible role of PBx for this purpose, and well-designed prospective studies should address this issue.

Overall, some important limitations may limit the use of a TBx + SBx strategy to selected, high-volume centers. A TBx + PBx approach needs a high-quality MRI reading, good fusion biopsy equipment and well-trained urologists, which may not always be available worldwide. Additionally, considering that nowadays a cognitive approach is considered to be equal to a fusion approach, how would we define perilesional sampling in cognitive biopsies? Moreover, our study is the first prospective study evaluating the role of perilesional sampling performed systematically, while the others are retrospective studies; therefore, more data are needed before definitive conclusions can be reached.

The introduction of mpMRI in clinical practice has improved the detection of csPCa. Although mpMRI screening offers numerous benefits in the clinical management of PCa, the rapid growth of acquisition methods and their widespread utilization have resulted in a significant variation in the quality of MRI scans obtained across different institutions, scanners and patients [6,25]. Multiple factors can contribute to the variability in the MRI quality, encompassing various aspects. These include the expertise of the MRI reader, time limitations that may compromise the signal-to-noise ratio, the specific equipment and software employed for imaging, the age of the scanner, pulse sequence parameters, the strength of the MRI magnet, the quality of surface coils, as well as patient-related factors like their body shape, motion, the presence of hip prostheses, rectal gas, dietary restrictions, and the use of antispasmodics or rectal enemas. To limit the possible bias related to the MRI quality, the Prostate Imaging Reporting and Data System (PI-RADS) has undergone several revisions (v.1 in 2012, v.2 in 2015, and v.2.1 in 2019) with the purpose of standardizing the acquisition and interpretation of prostate mpMRI [26]. The primary goal is to establish consistent guidelines for identifying abnormalities, including prostate cancer (PCa), on MRI scans and categorize the likelihood of these abnormalities having a clinical significance. Additionally, the Prostate Imaging Quality (PI-QUAL) scoring system was developed to evaluate the diagnostic quality of mpMRI based on criteria outlined in PI-RADS v.2, considering the T2-WI, DCE and DWI sequences [27]. While these efforts have undoubtedly improved the quality and uniformity of reporting, it is important to acknowledge that adhering to these standards does not guarantee an optimal quality in all cases. In fact, despite the implementation of PI-RADS guidelines, significant variations in quality persist among different centers worldwide.

In clinical practice, the analysis and interpretation of prostate mpMRI data currently depends on the expertise of human radiologists. However, despite their competence, these experts encounter limitations related to time, cost and scalability to meet the increasing demand for imaging. Moreover, the subjectivity of the PI-RADS core criteria combined with the challenging learning curve for interpreting prostate MRI leads to substantial variability among different observers. This variability is further compounded by the intricate and contentious nature of managing the high prevalence of prostate cancer, which exhibits diverse biological behaviors and overall prognoses [28,29].

The possible introduction of AI in the radiological field is an interesting area of research. With the abundance of large imaging datasets, there is a significant opportunity for the development of new algorithms and applications in the field of AI for prostate MRI. This advancement holds the potential to enhance clinical workflows and contribute to improved patient care. Prostate MRI is a well-suited modality for the application of AI due to its extensive use in lesion detection and its alignment with the current trend of utilizing targeted biopsy and local ablative therapies (LAT) to improve patients’ outcomes.

Recently, the EAU guidelines have recommended the trans-perineal route over the transrectal route when performing prostate biopsies. A recent meta-analysis of eight randomized clinical trials including 1596 patients compared both routes in terms of infectious complications [30]. The transrectal route resulted in a statistically significant higher risk of contracting infections when compared to that of the trans-perineal route (RR = 2.06 95% CI: 1.5–4.2) [30]. Such data clearly support the trans-perineal route over the transrectal route. However, the transperineal route may not be always available worldwide. Overall the trans-perineal route may need some sedation, and the learning curve is steeper when compared to that of the transrectal route [31,32,33]. In order to reduce the number of infectious complications related to transrectal biopsies, the use of a rectal povidone iodine preparation reduces the risk by 53% [30,34]. Overall, the rest of common complications, such as haematuria, haematospermia and urinary retention, do not differ between both routes.

The use of a PT + T strategy results in a lower number of cores per biopsy, which may result in lower infection rates and lower costs, specially in terms of pathological processing. In terms of infections, several researchers have assessed the impact of the number of cores on complication rates after prostate biopsies, confirming no association between the number of cores and complications [35,36]. Although our study does not include a cost analysis, reducing the number of cores can reduce our pathologists’ workload and the direct costs of PCa diagnosis. However, as physicians, we must diagnose significant PCa and balance the costs and risk of under-diagnoses. Future studies should evaluate the hypothesis that reducing the costs of PCa diagnosis by reducing the number of biopsies may result in higher cost due to delayed diagnoses.

The role of the patient in the decision-making process is always prominent, and clinicians should always consider this aspect in the management of patients at risk of PCa [37]. Patients should be carefully informed on the pros and cons of different approaches and may choose based on their preferences and expectations. Patients having a central role may probably improve the patient-reported outcomes in all the different stages of prostate cancer diagnosis and treatment [38]. An ongoing study by our group is analyzing the role of patient’s preferences and expectations in the diagnosis and treatment of PCa, and the results will be available soon.

Nowadays, several tools are available to increase the accuracy of PCa detection, and particularly, the detection of csPCa. Urine and serum biomarkers are commercially available to improve PCa detection. The PHI score, 4K score and IsoPSA have demonstrated good discrimination abilities with an AUC > 0.75 [39,40,41]. Urine biomarkers such as SelectMdx in combination with mpMRI resulted in a negative predictive value of 93% [20]. Additionally, in selected patients, the use of genetic classifiers can improve the detection of csPCa. Several authors have developed nomograms and calculators to integrate different variables and estimate the risks of PCa and csPCa [42,43]. These tools allow risk assessments to be performed, which can be deeply discussed with the patient to personalize the diagnostic approach.

In recent years, several authors have evaluated the possible role of artificial intelligence in prostate cancer diagnosis. AI is mainly based on machine learning algorithms. More specifically, machines learn representations and recognize patterns from training data. New data are then interpreted based on the training data in order to make decisions, such as classifications and predictions. AI models are an evolution of classic nomograms with fewer boundaries and the ability to evolve based on the input data [44,45].

Another interesting field is represented by nature language processors, such as ChatGPT. It represents the latest evolution of AI technology with multiple potential applications. Overall, ChatGPT has a huge computational capacity and may assist physicians in different tasks from diagnosis to treatment. The use of ChatGPT should not replace physicians, but should help physicians to enhance their precision and save time in many tasks. Such an instrument should be used with caution considering the legal issues and the sometimes biased and harmful results [46,47].

Prostate biopsies still carry an important risk of upgrading and downgrading in radical prostatectomy. Recently, the Gleason score was upgraded, and nowadays, the Epstein classification is used to grade radical prostatectomy [48]. The introduction of AI in histopathology is another interesting area of research. Eloy et al. recently evaluated the accuracy of ‘Paige Prostate’ in grading needle biopsies. The use of AI increased the accuracy of diagnosis by pathologists, reduced the time needed by 20% and reduced the number of immunohistochemistry procedures needed [49]. In the future, the implementation of such AI models will help with the standardization of histopathological reporting.

The use of AI probably represents the future of PCa diagnosis. Future applications may include the implementation of AI in the fusion biopsy machines to aid the fusion phase and improve core tracking. Most of the fusion machines available may not adapt to the modifications of the prostate due to the compression of the probe, local anesthesia and the biopsy cores taken. For all these reasons, nowadays, there are very few differences between cognitive, fusion and in bore techniques. Biopsy results clearly influence prostate cancer management, and therefore, every effort should be made to grade and stage PCa adequately.

Finally, in recent years, the introduction of PET/CT has improved the diagnosis of recurrent PCa. Some authors have proposed its used in PCa diagnosis. In 2022, Qiu et al. performed a pilot study to evaluate the possible role of PET/CT in this setting. According to their results, dual-tracer PET/CT screens out patients for avoiding 52.67% (59/112) of the unnecessary biopsy, whereas dual-tracer PET/CT-TB plus SB achieved a high detection rate (77.36%), without the misdiagnosis of csPCa. Although this represents an interesting area of research, it cost and availability still represent an important limitation [50]. More data are needed before definitive conclusions can be reached.

We have to acknowledge some limitations of the present study. This study was conducted on a consecutive series of patients undergoing transrectal, free hand fusion biopsies; therefore, our results may not be generalized to other cohorts using different approaches. However, to our knowledge this is the first prospective series analyzing the role of perilesional sampling in PCa diagnosis. The sample size may be considered as a limitation of the study, particularly considering the number of PIRADS 5 patients enrolled. However, a dedicated study is ongoing to evaluate this strategy in this particular category of patients at high risk of PCa. In our study, we excluded patients with PSA > 20 ng/mL and previous negative biopsies, which may be considered as a limitation of the study. However, considering the higher and lower risks of csPCa of these groups of patients, respectively, we preferred to exclude these patients to avoid further biases in the analysis. Moreover, according to our department protocol, MRI is not routinely performed on patients with a very high risk of PCa, such as patients with an elevated PSA (>20 ng/L) and positive DRE. Our strategy is related to the long waiting list for MRI and the need for quick diagnoses in these patients.

Finally, the study presents the common limitation of biopsy cohorts with no final pathology available. A study to evaluate the role of perilesional sampling on upgrading at the final pathology stage is ongoing, and the results will be available soon.

## 5. Conclusions

PCa diagnosis is still challenged by the overdiagnosis of not significant cancers and the underdiagnosis of significant cancers. Many efforts have been made since the introduction of mpMRI to improve the detection of csPCa. However, the number of insignificant cancers is still very high. The median number of biopsies per patient has dramatically increased, and consequently, there are higher costs and workloads for pathologists. According to our results, a biopsy strategy including targeted and peri-targeted biopsies is clearly suboptimal in the detection of csPCa. Missing 7% of csPCa is probably not acceptable nowadays. Notwithstanding the advantages in terms of the reduction of nsPCa diagnoses and number of cores, the umbra and penumbra strategy is still far from useable. Clinicians should focus in improving the selection of patients needing prostate biopsies. Obtaining MRI readings using AI models and ChatGPT is probably the path to follow.

## Figures and Tables

**Figure 1 life-13-01719-f001:**
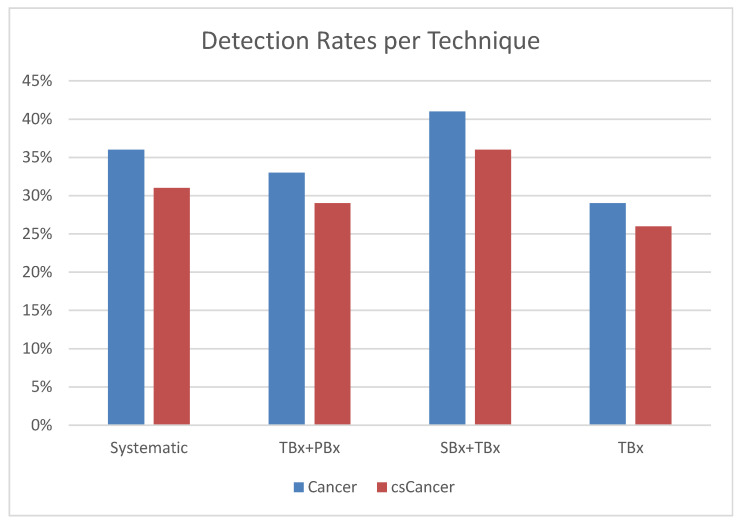
Detection of Cancer and csCancer in different biopsy strategies.

**Table 1 life-13-01719-t001:** Baseline characteristics of the cohort population.

Patients	262
Age (years)	70 (64/76); 70 ± 7.8
PSA (ng/mL)	6.8 (4.3/10.5); 8.2 ± 6.4
BMI (Kg/m^2^)	26.5 (23.9/29); 26.4 ± 4.4
PSAdensity (ng/mll/cc)	0.14 (0.1/0.2); 0.9 ± 3.4
Prostate Volume (cc)	52 (36/67.7); 58.7 ± 34.6
Systematic Cores	12 (12/12); 12 ± 0
Targeted Cores	3(3/6); 4.3 ± 1
Perilesional cores	3(3/6); 4.5 ± 0.8
PIRADS TARGET 1	262
3	99/262: 38%
4	128/262: 49%
5	34/262: 13%
PIRADS TARGET 2	66
3	20/66: 30%
4	32/66: 49%
5	14/66: 21%
ISUP Grade	
0	148/262: 57%
1	13/262: 7%
2	16/262: 6%
3	21/262: 8%
4	21/262: 8%
5	37/262: 14%

**Table 2 life-13-01719-t002:** Cancer detection rates.

Technique	Cancer	Clinically Significant Cancer
Systematic + Target + Peri-target	43.5% (114/262)	35% (92/262)
Systematic Biopsies	35.9% (94/262)	30.5% (80/262)
Systematic + Target	40.5% (106/262)	35% (92/262)
Target + Peri-target	32.8% (86/262)	29% (76/262)
Target	29% (76/262)	26% (68/262)

**Table 3 life-13-01719-t003:** Missed clinically significant cancer and non-significant cancers using different techniques.

	Missed csCancer	Missed nsCancer	Diagnosed csCancer	Diagnosed nsCancer
Systematic Biopsies	10/92 (11%)	8/20 (80%)	82/92 (89%)	12/20 (60%)
Systematic + Target	2/92 (2%)	4/20 (20%)	90/92 (98%)	16/20 (80%)
Target + Peri-target	18/92 (20%)	8/20 (40%)	74/92 (80%)	12/20 (60%)
Target	22/92 (24%)	14/20 (70%)	70/92 (76%)	6/20 (30%)
Total	92	20	92	20

**Table 4 life-13-01719-t004:** Sub-analysis for PIRADS 4–5.

	Missed csCancer	Missed nsCancer	Diagnosed csCancer	Diagnosed nsCancer
Systematic Biopsies	10/90 (11%)	6/16 (37%)	80/90 (89%)	10/16 (63%)
Systematic + Target	2/90 (2%)	2/16 (13%)	88/90 (98%)	14/16 (87%)
Target + Peri-target	8/90 (9%)	6/16 (37%)	82/90 (91%)	10/16 (63%)
Target	14/90 (16%)	10/16 (62%)	76/90 (84%)	6/16 (38%)
Total	90	16	90	20

**Table 5 life-13-01719-t005:** Sub-analysis for PIRADS 5.

	Missed csCancer	Missed nsCancer	Diagnosed csCancer	Diagnosed nsCancer
Systematic Biopsies	0/20 (0%)	1/5 (20%)	20/20 (100%)	4/5 (80%)
Systematic + Target	0/20 (0%)	0/5 (13%)	20/20 (100%)	5/5 (100%)
Target + Peri-target	2/20 (10%)	2/5 (40%)	18/20 (90%)	3/5 (60%)
Target	2/20 (10%)	2/5 (40%)	18/20 (90%)	3/5 (60%)
Total	20	5	20	20

## Data Availability

Data will be available upon request.

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
