# Peer review of "Role of Perilesional Sampling of Patients Undergoing Fusion Prostate Biopsies"

_life, 2023, doi:10.3390/life13081719_

Round 1
Reviewer 1 Report
Perilesional sampling during prostate biopsies can be considered as an alternative or complementary approach to systematic biopsies in patients at risk of prostate cancer. This technique involves targeting specific suspicious areas or lesions within the prostate gland for biopsy, rather than sampling the entire gland systematically. In patients at risk of prostate cancer, such as those with elevated prostate-specific antigen (PSA) levels or abnormal findings on imaging studies, perilesional sampling can offer several potential advantages.
Over all it’s a good study and I am in the favor of its acceptance for the publication in the journal.
Just a few suggestions and recommendations on the study are as follows:
Perilesional sampling during prostate biopsies refers to the targeted sampling of tissue around a suspicious lesion or abnormality in the prostate gland. The technique is used to obtain additional tissue samples from areas that are more likely to contain cancer cells, thus potentially reducing the need for systematic biopsies.
By targeting specific areas of interest, perilesional sampling may increase the likelihood of detecting prostate cancer if it is present in a particular lesion [https://doi.org/10.1016/j.eururo.2022.01.008].
This approach can be particularly useful in patients with known risk factors for prostate cancer or in cases where imaging studies indicate suspicious areas.
Systematic biopsies, on the other hand, involve the sampling of multiple areas of the prostate gland in a systematic pattern, regardless of the presence of any visible lesions. This approach aims to detect cancerous cells that may be present in different regions of the prostate.
The authors are suggested to describe the comprehensive differentiating features of both [Perilesional and Systematic] to establish the rationale of the study that is not well presented in this study.
The decision to perform perilesional sampling instead of or in addition to systematic biopsies depends on several factors, including the individual patient's risk profile, clinical findings, imaging results, and the Clinical judgment.
Perilesional sampling alone may not be sufficient for a comprehensive evaluation of the entire prostate gland. In some cases, systematic biopsies may still be necessary to ensure a more thorough assessment and to detect any cancerous cells that might be present in regions not targeted by the perilesional sampling.
Clinically significant prostate cancer is typically determined based on a combination of factors, including the tumor grade (Gleason score), tumor volume, and the extent of cancer spread beyond the prostate gland. High-grade tumors (Gleason score 7 or higher) with a large volume and/or evidence of extraprostatic extension are generally considered clinically significant.
The authors are suggested, to give a comparison on the basis of other factors too.
It is suggested that, the adopted statistical analyses methods should be clearly described separately under materials and method section.
Author Response
- Perilesional sampling during prostate biopsies can be considered as an alternative or complementary approach to systematic biopsies in patients at risk of prostate cancer. This technique involves targeting specific suspicious areas or lesions within the prostate gland for biopsy, rather than sampling the entire gland systematically. In patients at risk of prostate cancer, such as those with elevated prostate-specific antigen (PSA) levels or abnormal findings on imaging studies, perilesional sampling can offer several potential advantages.
Overall it’s a good study and I am in the favor of its acceptance for the publication in the journal.
Just a few suggestions and recommendations on the study are as follows:
We thank the reviewer for taking time to review our manuscript and for the valuable suggestions.
- Perilesional sampling during prostate biopsies refers to the targeted sampling of tissue around a suspicious lesion or abnormality in the prostate gland. The technique is used to obtain additional tissue samples from areas that are more likely to contain cancer cells, thus potentially reducing the need for systematic biopsies.
By targeting specific areas of interest, perilesional sampling may increase the likelihood of detecting prostate cancer if it is present in a particular lesion [https://doi.org/10.1016/j.eururo.2022.01.008].
This approach can be particularly useful in patients with known risk factors for prostate cancer or in cases where imaging studies indicate suspicious areas.
Systematic biopsies, on the other hand, involve the sampling of multiple areas of the prostate gland in a systematic pattern, regardless of the presence of any visible lesions. This approach aims to detect cancerous cells that may be present in different regions of the prostate.
The authors are suggested to describe the comprehensive differentiating features of both [Perilesional and Systematic] to establish the rationale of the study that is not well presented in this study.
We thank the reviewer for the suggestions. Introduction has been improved as suggested:
Systematic biopsies refer to the sampling of 5-6 random cores per prostate lobe and was considered the standard of care for many years before the introduction of mpMRI. Such a strategy has the aim of sampling the different areas of the prostate. Target biopsies refers to the sampling of the mpMRI index lesion with 3-6 cores with a cognitive or fusion approach. Peri-targeted biopsies are defined as cores sampling the surrounding area of the lesion within different radius from the lesion. Such a strategy may detect cancer cells in the surrounding area of the lesion (Brisbane et al).
- The decision to perform perilesional sampling instead of or in addition to systematic biopsies depends on several factors, including the individual patient's risk profile, clinical findings, imaging results, and the Clinical judgment.
Perilesional sampling alone may not be sufficient for a comprehensive evaluation of the entire prostate gland. In some cases, systematic biopsies may still be necessary to ensure a more thorough assessment and to detect any cancerous cells that might be present in regions not targeted by the perilesional sampling.
Clinically significant prostate cancer is typically determined based on a combination of factors, including the tumor grade (Gleason score), tumor volume, and the extent of cancer spread beyond the prostate gland. High-grade tumors (Gleason score 7 or higher) with a large volume and/or evidence of extra-prostatic extension are generally considered clinically significant.
The authors are suggested, to give a comparison on the basis of other factors too.
We thank the reviewer for his/her comment and for the possibility to improve our manuscript. In our manuscript we have evaluated the role of perilesional sampling in all patients and dividing patients by different PIRADS score confirming that perilesional sampling is suboptimal for the diagnosis of csPCa. As suggested by the reviewer we have performed a separata analysis on the basis of other factors such as age ( <70 vs >70) and by PSAdensity ( <0,15 vs >0,15) which are well known predictors of PCa. Discussion section has been updated commenting on these results to improve the manuscript.
Discussion section:
A sub-analysis to evaluate the role of peri-lesional sampling in patients older than 70 years old and in patients with a PSA density>0,15 was performed (data not shown). In both groups of patients younger and older than 70 years, the T+PT strategy missed 5% and 6% of clinically significant cancer respectively (p>0,05). As well in both group of patients with PSA density >0,15 or >0,15, the T+PT strategy missed 7% and 7% of clinically significant cancer resspectively (p>0,05). This sub-analysis confirms the fact that the T+PT strategy is to be considered suboptimal for the diagnosis of PCa even in different subgroups of patients.
- It is suggested that, the adopted statistical analyses methods should be clearly described separately under materials and method section.
We thank the reviewer for his/her comment and for the possibility to improve our manuscript. The statistical analysis part has been divided and improved.
See New Statistical analysis paragraph :
2.1 Statistical Analysis
Categorical variables were presented as frequency distributions and percentages, while continuous variables were represented by medians along with interquartile ranges (IQRs). To assess the detection rates of not significant prostate cancer (nsPCa) and clinically significant prostate cancer (csPCa), the approaches of targeted biopsy (TBx) and TBx with perilesional biopsy (PBx) were compared to the detection achieved by TBx with systematic SBx, considered as the "gold standard." Sensitivities and their corresponding 95% confidence intervals (CIs) were calculated using binomial tests. Statistical significance was determined by non-overlapping 95% CIs between sensitivity values.
A subgroup analysis was conducted based on the age, PSA density and PI-RADS assessment category. Differences in detection rates between subgroups were evaluated using the Fisher's exact test with the Freeman-Halton extension. All statistical analyses were performed using SPSS for MacOS, version 27 (IBM, Armonk, NY, USA).
Reviewer 2 Report
Thank you for your interesting paper!
My first suggestion is that you clarify a bit your exclusion criteria; PSA above 20 ng does not exclude the indication for biopsy, although the need for fusion is debatable. i consider this should be motivated in the paper. Same for previous negative biopsies, as this is actually a better indication for the fusion technique.
I suggest you have a look and cite the following paper: doi:http://dx.doi.org/10.11152/mu-2932
Just as a suggestion, I would add a short paragraph about the impact of taking more biopsies on the infection risk and maybe even the cost of the procedure.
Author Response
Thank you for your interesting paper!
My first suggestion is that you clarify a bit your exclusion criteria; PSA above 20 ng does not exclude the indication for biopsy, although the need for fusion is debatable. i consider this should be motivated in the paper. Same for previous negative biopsies, as this is actually a better indication for the fusion technique.
We thank the reviewer for his/her comment. We agree that the decision to exclude these categories of patients is debatable. In our study patients with a PSA>20ng/ml were excluded due to the high probability of cancer while those with previous negative biopsies to avoid biases due to the lower detection rate in this group of patients. According to our department protocol MRI is not routinely performed in patients with a very high-risk of PCa such as patients with elevated PSA ( >20ng/l) and positive DRE. Our strategy is related to the long waiting list for an MRI and the need of a quick diagnosis in these patients. The present is now better specify in the limitation section.
See Discussion section:
In our study we excluded patients with PSA>20 ng/ml and previous negative biopsies which may be considered a limitation of the study. However, considering the higher and lower risk of csPCa of these groups of patients respectively we preferred to exclude these patients to avoid further biases in the analysis. Moreover, according to our department protocol, MRI is not routinely performed in patients with a very high-risk of PCa such as patients with elevated PSA ( >20ng/l) and positive DRE. Our strategy is related to the long waiting list for an MRI and the need of a quick diagnosis in these patients.
I suggest you have a look and cite the following paper: doi:http://dx.doi.org/10.11152/mu-2932
Just as a suggestion, I would add a short paragraph about the impact of taking more biopsies on the infection risk and maybe even the cost of the procedure.
We thank the reviewer for his/her comment and for the possibility to improve our manuscript.
The reference has been added and a short paragraph has been added to the discussion section.
See new Reference 9.
See Discussion section.
The use of a PT + T strategy results in a lower number of cores per biopsy which may result in lower infection rates and lower cost specially in terms of pathological processing. In terms of infections several papers have assessed the impact of the number of cores on complication rates after prostate biopsies confirming no association between the number of cores and complications (Loeb 2012, Huang 2016). Although our study does not allow a cost analysis, reducing the number of cores can reduce our pathologist workload and direct costs of PCa diagnosis. However, as physicians we must diagnose significant PCa and balance costs and risk of under-diagnosis. Future studies should evaluate the hypothesis that reducing the costs of PCa diagnosis by reducing the number of biopsies may result in higher cost due to delayed diagnosis.
Round 2
Reviewer 2 Report
Thank you for making the changes I suggested.